# A CAM-Related NF-YB Transcription Factor Enhances Multiple Abiotic Stress Tolerance in *Arabidopsis*

**DOI:** 10.3390/ijms25137107

**Published:** 2024-06-28

**Authors:** Naleeka R. Malwattage, Beate Wone, Bernard W. M. Wone

**Affiliations:** Department of Biology, University of South Dakota, Vermillion, SD 57069, USA

**Keywords:** nuclear factor Y, transcription factor, abiotic stress, Crassulacean acid metabolism, climate change, extremophytes

## Abstract

Abiotic stresses often occur simultaneously, and the tolerance mechanisms of plants to combined multiple abiotic stresses remain poorly studied. Extremophytes, adapted to abiotic stressors, might possess stress-adaptive or -responsive regulators that could enhance multiple abiotic stress resistance in crop plants. We identified an NF-YB transcription factor (TF) from the heat-tolerant obligate Crassulacean acid metabolism (CAM) plant, *Kalanchoe fedtschenkoi*, as a potential regulator of multiple abiotic stresses. The *KfNF-YB3* gene was overexpressed in *Arabidopsis* to determine its role in multiple abiotic stress responses. Transgenic lines exhibited accelerated flowering time, increased biomass, larger rosette size, higher seed yield, and more leaves. Transgenic lines had higher germination rates under combined NaCl, osmotic, and water-deficit stress treatments compared to control plants. They also showed enhanced root growth and survival under simultaneous NaCl, osmotic, water-deficit, and heat stress conditions in vitro. Interestingly, potted transgenic lines had higher survival rates, yield, and biomass under simultaneous heat, water-deficit, and light stresses compared to control plants. Altogether, these results provide initial insights into the functions of a CAM-related TF and its potential roles in regulating multiple abiotic stress responses. The CAM abiotic stress-responsive TF-based approach appears to be an ideal strategy to enhance multi-stress resilience in crop plants.

## 1. Introduction

Global crop production is expected to decline due to increasing abiotic stressors, including extremes in temperature, water availability, light intensity, and nutrient levels, exacerbated by climate change and anthropogenic activities [1,2,3]. These stressors will significantly affect C_3_ crop plants, negatively impacting their growth and productivity [4]. The stability of plant growth and survival decreases significantly as the number and complexity of stresses increase simultaneously [5]. Studies have shown that multiple abiotic stresses can have additive effects, with the simultaneous occurrence of two or more stresses creating a synergistic detrimental effect that leads to severe and irreversible damage [5,6].

More importantly, abiotic stresses often occur simultaneously, and the tolerance mechanisms in plants to combined multiple abiotic stresses remain poorly studied [7,8,9]. Yan et al. [5] recently reported that common combinations of multiple abiotic stresses in horticultural crops include heat and drought stress, heat–drought–light and salt stress, heat and waterlogging stress, and heat and salinity stress. Understanding the effects of abiotic stress combinations in plants is crucial for developing more climate change-resilient crop plants [1,10]. Developing multi-stress-resilient crop varieties is emerging as the most sustainable solution for ensuring global food security under climate change-induced weather events, such as extreme temperatures and prolonged or flash droughts [1,5,11].

A straightforward approach to improving multiple abiotic stress resistance in plants could involve utilizing abiotic stress-adaptive or -responsive transcription factors (TFs) from extremophytes to enhance tolerance in crop plants [12,13,14]. Extremophytes are adapted to multiple abiotic stressors because they possess various abiotic stress-adaptive or -responsive regulators [11,14,15,16]. Here, we provide initial insights into the multiple abiotic stress response function of a transcription factor from the heat-tolerant obligate Crassulacean acid metabolism (CAM) plant *Kalanchoe fedtschenkoi*. Understanding the role of potential CAM-related abiotic stress-responsive TFs could offer an effective strategy for bioengineering desirable responses to multiple abiotic stress tolerances in non-CAM plants.

CAM plants are well-adapted to water-limited environments due to their higher water-use efficiency (WUE) and abiotic stress resistance compared to C_3_ plants [17,18,19]. We have identified an NF-YB transcription factor (TF) from the heat-tolerant obligate CAM plant *Kalanchoe fedtschenkoi* as a potential regulator of multiple abiotic stresses [14]. Notably, this candidate *NF-YB3* gene exhibited expression levels fourfold higher than those observed in the C_3_ state during CAM induction in older leaf pairs of *K. fedtschenkoi* [14].

The NF-Y is a ubiquitous plant transcription factor expressed as one of three possible isoforms: NF-YA, NF-YB, or NF-YC [20]. These isoforms interact to form a heterotrimeric complex, with each subunit playing key roles in responses to abiotic stresses, including drought, heat, freezing, and salinity [20,21]. Despite its potential importance, the specific role of *KfNF-YB3* in abiotic stress response remains unclear. To gain initial insight into the function of the *KfNF-YB3* gene, we generated 35S::*KfNF-YB3* overexpression lines in *Arabidopsis*. Our results indicated that increased expression of *KfNF-YB3* leads to earlier flowering, greater rosette size, higher plant height, and increased seed yield under combined multiple abiotic stressors compared to control *Arabidopsis*.

## 2. Results

### 2.1. Orthologues of KfNFYB3 TF in Arabidopsis thaliana

The closest orthologue, *AtNF-YB3* (AT4G14540), showed 81% sequence identity with *KfNF-YB3* (Appendix A). *AtNF-YB3* is expressed in the flowering, cotyledon, and mature leaf stages. These results help predict the expression pattern of the *KfNF-YB3* transcription factor in transgenic *Arabidopsis* lines. Furthermore, this orthologue has been functionally characterized for its role in accelerating flowering time in *Arabidopsis* yet has not been reported for its roles in abiotic stress tolerance.

The alignment with several abiotic-responsive NF-YB transcription factor subunits from different plant species showed a highly conserved region (~74 amino acid residues) within the KfNF-YB3 sequence, exhibiting strong similarity to the NF-YB subunits from various plant species (Figure 1A,B). Also, the phylogeny of *KfNF-YB3* showed recent homology with *PwNF-YB3* from *Picea wilsonii* (Figure 1C).

### 2.2. KfNF-YB3 Expression in Overexpressing Lines

Relative quantification (RQ) of four selected *KfNF-YB3* transgenic lines, *OxKfNFY-1*, *OxKfNFY-2*, *OxKfNFY-16*, and *OxKfNFY-17*, resulted in 14.9, 308.5, 76.9, and 123.6, respectively. These four lines were used for downstream phenotyping and abiotic stress assays and are referred to as *KfNFY-YB3* transgenic lines from here on.

### 2.3. Growth, Development, and Flowering Time

The overexpression of *KfNFY-B3* increased rosette size, number of leaves, height, shoot biomass, and seed yield in transgenic *Arabidopsis* lines. After germination in full-strength MS media, a significant morphological difference was observed between the 14-day-old Col-0 WT control and transgenic seedlings (Figure 2A). The rosette diameters of transgenic lines *OxKfNFY-1*, *OxKfNFY-2*, *OxKfNFY-16*, and *OxKfNFY-17* were 25.6 ± 1.06 mm, 23.9 ± 1.86 mm, 22.3 ± 1.76 mm, and 22.0 ± 1.41 mm, respectively, whereas Col-0 WT control plants had an average rosette diameter of 13.0 ± 1.7 mm after 2 weeks (Figure 2B, *F* = 7.09, *p* < 0.01).

After five weeks, the rosette sizes of transgenic lines were significantly larger compared to Col-0 WT control plants (Figure 2D). The average rosette diameters of the four transgenic lines were 10.3 ± 0.39 cm, 10.9 ± 0.89 cm, 7.6 ± 0.67 cm, and 9.5 ± 1.06 cm, whereas Col-0 WT control plants had an average diameter of 0.1 ± 0.08 cm (*F* = 5.69, *p* < 0.01).

The heights of three transgenic lines, *OxKfNFY-1*, *OxKfNFY-16*, and *OxKfNFY-17*, were significantly different compared to Col-0 WT after five weeks (Figure 3A). However, the transgenic line *OxKfNFY-2* did not show a significant difference in plant height compared to Col-0 WT (*n* = 30, *F* = 10.32, *p* = 0.16). Transgenic lines had significantly more leaves compared to Col-0 WT control plants (Figure 3B). The numbers of leaves for transgenic lines *OxKfNFY-1*, *OxKfNFY-2*, *OxKfNFY-16*, and *OxKfNFY-17* were 19.3 ± 1.56, 17 ± 0.87, 18.2 ± 0.66, and 18.7 ± 1.11, respectively, whereas Col-0 WT control plants had significantly fewer leaves at 8.3 ± 1.37 (*F* = 8.54, *p* < 0.01).

Shoot dry weight was measured after seven weeks. All transgenic lines had significantly greater shoot dry weight/biomass (*F* = 11.89, *p* < 0.01) compared to Col-0 WT control plants (Figure 3C). The shoot dry weights of *OxKfNFY-1*, *OxKfNFY-2*, *OxKfNFY-16*, and *OxKfNFY-17* were 0.4 ± 0.04 g, 0.34 ± 0.03 g, 0.4 ± 0.04 g, and 0.4 ± 0.03 g, respectively, whereas Col-0 WT control had 0.1 ± 0.04 g after five weeks. Furthermore, the transgenic lines *OxKfNFY-1*, *OxKfNFY-2*, *OxKfNFY-16*, and *OxKfNFY-17* showed significantly higher seed yields of 0.11 ± 0.01 g, 0.14 ± 0.01 g, 0.1 ± 0.01 g, and 0.1 ± 0.01 g, respectively, whereas the Col-0 WT control had 0.06 ± 0.01 g (Figure 3D, *F* = 5.32, *p* < 0.01).

*KfNFY-B3* overexpression shortens flowering time in transgenic *Arabidopsis* lines (Figure 3E). Under long-day (LD) photoperiod conditions, the numbers of days to visible first flower buds were recorded for four transgenic lines and WT control plants. The average numbers of days to the first flower buds were 23.6 ± 1.69, 23.3 ± 1.54, 23.7 ± 1.73, and 25.1 ± 1.42 for transgenic lines *OxKfNY-1*, *OxKfNY-2*, *OxKfNY-16*, and *OxKfNY-17*, respectively. These average days were significantly shorter compared to the Col-0 WT control plants, which had an average of 32.8 ± 1.63 days (*F* = 1.11, *p* = 0.40).

### 2.4. Integrated Water-Use Efficiency

Overexpression of *KfNFY-B3* leads to increased water-use efficiency (WUE) in transgenic *Arabidopsis* lines. The WUE experiment was conducted in a closed system for four weeks under long-day (LD, 16 h light/8 h dark) conditions (Figure 4A). Transgenic lines exhibited significantly higher WUE compared to the Col-0 WT plants (*n* = 30, *F* = 1.67, *p* < 0.01). The water-use efficiencies of the four transgenic lines, *OxKfNFY-1*, *OxKfNFY-2*, *OxKfNFY-16*, and *OxKfNFY-17*, were 0.1 ± 0.01, 0.1 ± 0.01, 0.1 ± 0.004, and 0.1 ± 0.01 g/mL of water used, respectively, whereas Col-0 WT plants had an average WUE of 0.02 ± 0.01 g/mL of water used (Figure 4B).

### 2.5. Water-Deficit Stress Tolerance

Transgenic *KfNFY-B3 Arabidopsis* lines have enhanced tolerance to water-deficit stress treatment (Figure 5A,B). The survival rates of transgenics and Col-0 WT controls were assessed after four days of re-watering. The survival percentage of WT seedlings was 0%, whereas *OxKfNFY-1*, *OxKfNFY-2*, *OxKfNFY-16*, and *OxKfNFY-17* were 83.3 ± 8.82, 73.3 ± 8.82, 90.0, and 93.3 ± 3.33% (Figure 5C).

All transgenic lines had increased shoot dry weight/biomass after re-watering compared to the Col-0 WT control plants (Figure 5D). Shoot dry weights (g) of *OxKfNFY-1*, *OxKfNFY-2*, *OxKfNFY-16*, and *OxKfNFY-17* were 0.4 ± 0.04, 0.4 ± 0.03, 0.9 ± 0.04, and 1.0 ± 0.12 g, respectively, whereas those of the Col-0 WT control were 0.1 ± 0.03, 0.2 ± 0.03, 0.2 ± 0.03, and 0.2 ± 0.02 g after four weeks. Furthermore, transgenic lines *OxKfNFY-1*, *OxKfNFY-2*, *OxKfNFY-16*, and *OxKfNFY-17* showed higher seed yields (g) of 0.1 ± 0.01, 0.1 ± 0.01, 0.07 ± 0.01, and 0.08 ± 0.01, respectively, whereas those of the Col-0 WT control plants were 0.01 ± 0.003, 0.01 ± 0.004, 0.02 ± 0.005, and 0.01 ± 0.004 after drying for 48 h at 70 °C (Figure 5E).

### 2.6. Hypocotyl Length under Heat Stress

Transgenic *KfNFY-B3* lines had elongated hypocotyl length under heat stress treatment (Figure 6A). Transgenic lines had significantly longer hypocotyls compared to WT control plants (*n* = 30, *F* = 2.24, *p* <0.01). Hypocotyl elongation lengths of four transgenic lines, *OxKfNFY-1*, *OxKfNFY-2*, *OxKfNFY-16*, and *OxKfNFY-17* in MS medium supplemented with 200 mM NaCl + −0.5 MPa + PEG were 3.2 ± 0.4, 3.7 ± 0.62, 3.6 ± 0.2, and 2.3 ± 0.3 mm, respectively, whereas the Col-0 WT control was not elongated (Figure 6B–E).

### 2.7. Single and Combined Multiple Abiotic Stress Tolerance on Germination

The overexpression of *KfNF-YB3* in transgenic *Arabidopsis* demonstrated positive responses to individual and combined abiotic stressors, including mannitol (as the osmoticum), sodium chloride (NaCl), polyethylene glycol (PEG), and water-deficit conditions, during both germination and post-germination phases (Figure 7). The germination rates of transgenic lines were significantly higher than those of Col-0 WT plants under all stress treatments (Figure 7B–G).

In the post-germination combined stress assay involving NaCl and osmotic stress, primary root elongation was measured after 12 days of seedling growth in MS media containing 200 mM NaCl and PEG-8000. Under these combined stress conditions, transgenic lines exhibited significantly longer primary root lengths compared to Col-0 WT plants (Figure 7I). The primary root lengths of *OxKfNFY-1*, *OxKfNFY-2*, *OxKfNFY-16*, and *OxKfNFY-17* were 74.0 ± 2.41 mm, 74.4 ± 2.30 mm, 64.6 ± 1.47 mm, and 65.8 ± 0.95 mm, respectively, while the Col-0 WT control root length was 27.7 ± 2.27 mm after 12 days (*F* = 4.24, *p* < 0.01).

Furthermore, transgenic lines *OxKfNFY-1*, *OxKfNFY-2*, *OxKfNFY-16*, and *OxKfNFY-17* exhibited higher shoot dry weights of 4.2 ± 0.41 mg, 5.8 ± 0.81 mg, 4.9 ± 0.48 mg, and 6.3 ± 0.91 mg, respectively, compared to 1.2 ± 0.45 mg in Col-0 WT plants after 48 hours of drying at 70 °C (Figure 7J, *F* = 6.54, *p* < 0.01).

### 2.8. Combined Multiple Abiotic Stress Tolerance in Transgenic Arabidopsis Seedlings

Overexpression of *KfNF-YB3* resulted in a higher survival rate under combined osmotic, NaCl, water-deficit, and heat stress (Figure 8A). Survival rates of transgenic seedlings were higher than those of Col-0 WT control seedlings in all stress treatments (Figure 8B–G).

### 2.9. Combined Multiple Abiotic Stress Tolerance in Potted Transgenic Arabidopsis

Overexpression of *KfNFY-B3* in transgenic *Arabidopsis* enhanced tolerance to simultaneous water-deficit, heat, and light stress (Figure 9A,B). Significant differences in growth rates were observed between the transgenic lines and Col-0 WT control plants, with transgenic lines exhibiting more shoots and siliques.

Survival rates of transgenic plants under combined multiple stress treatment were higher than those of Col-0 WT control plants (Figure 9C). The survival rates of *OxKfNFY-1*, *OxKfNFY-2*, *OxKfNFY-16*, and *OxKfNFY-17* were 90%, 93.34%, 90%, and 96.67%, respectively, compared to less than 5% for the Col-0 WT control. Additionally, shoot dry weight was measured after re-watering. Transgenic lines showed greater shoot dry weight/biomass compared to Col-0 WT control plants (Figure 9D). The shoot dry weights of *OxKfNFY-1*, *OxKfNFY-2*, *OxKfNFY-16*, and *OxKfNFY-17* were 0.507 ± 0.0421, 0.446 ± 0.044, 0.897 ± 0.049, and 0.998 ± 0.033 g, respectively, while Col-0 WT control values were 0.119 ± 0.026, 0.171 ± 0.031, 0.181 ± 0.025, and 0.167 ± 0.023 g after stress treatment. Furthermore, transgenic lines *OxKfNFY-1*, *OxKfNFY-2*, *OxKfNFY-16*, and *OxKfNFY-17* exhibited significantly higher seed yields of 0.04 ± 0.002, 0.04 ± 0.0024, 0.05 ± 0.003, and 0.03 ± 0.003 g, respectively, compared to the Col-0 WT control seed yields of 0.003 ± 0.0006, 0.004 ± 0.008, 0.006 ± 0.0018, and 0.005 ± 0.0015 g (Figure 9E).

## 3. Discussion

To our knowledge, this study is the first to describe the potential role of an abiotic stress-responsive transcription factor from a heat-tolerant obligate CAM plant, *Kalanchoe fedtschenkoi*, in enhancing multiple plant abiotic stress tolerance, WUE, and plant growth and development. Our collective results found that overexpression of *KfNF-YB3* in *Arabidopsis* enhanced survival rate, shoot biomass, and seed yield under combined multiple abiotic stresses, including water-deficit, NaCl, osmotic, heat, and high-irradiance stresses. Drought and heat stress represent excellent examples of two different abiotic stress conditions that occur simultaneously in the field [5,22,23]. High temperatures and drought stress can also frequently be combined with high irradiance and high salinity [5,24,25]. Several studies have examined the effects of a combination of drought and heat stress on the growth and productivity of maize, barley, sorghum, and various grasses [26,27]. These studies found that a combination of drought and heat stress had a significantly greater detrimental effect on the growth and productivity of these plants and crops compared to the individual stresses applied separately.

Previous studies have revealed the role of NF-Y transcription factors in the responses to individual abiotic stresses, such as drought, salt, cold, and heat. Many NF-YA, NF-YB, and NF-YC subunits play key roles in the drought response [28,29]. Overexpressing *AtNF-YA5* reduced leaf water loss and increased resistance to drought stress compared to the wild type, whereas *NF-YA5* mutants were more sensitive to drought stress, suggesting that NF-YA5 positively regulates drought stress responses [29].

In addition to NF-YA, NF-YB is also involved in *Arabidopsis* drought stress tolerance. Overexpression of *AtNF-YB1* enhanced plant drought resistance [29,30]. Furthermore, the NF-YA2-B3-C10 ternary complex enhanced the expression of the heat stress-inducible gene *HEAT SHOCK FACTOR A3* (*HsfA3*) during heat stress responses in cooperation with *DREB2A* [31]. Although extensive genetic evidence indicates that many individual NF-Y subunits function in abiotic stresses, most studies focus on individual stresses. The role of NF-Ys in abiotic stress combinations remains elusive [20]. Interestingly, the NF-Y is considered as a master transcription factor in the desiccation tolerance response in the anhydrobiotic larvae of the African midge, *Polypedilum vanderplanki* [32]. This suggests that the regulatory mechanisms of abiotic stress resistance are conserved in both plants and animals. Whether the NF-Y is a master regulator of multiple abiotic stresses or not in plants needs to be further explored.

According to Singh and Laxmi [33], abscisic acid (ABA) modulates the activity of NF-Y transcription factors. Previous studies have demonstrated a direct and indirect involvement of NF-Ys in ABA-regulated post-germination pathways under abiotic stresses such as drought and salinity [20,34]. Therefore, the possible mechanism of combined abiotic stress response by *KfNF-YB3* TF could involve the interaction of NF-YB with ABA in the transgenic plants. Further analysis of ABA accumulation and exogenous application in transgenic lines under abiotic stress would confirm their roles in the stress response mechanism.

*KfNF-YB3* TF overexpression improved plant performance under multiple abiotic stress conditions. Plants have evolved a variety of regulatory mechanisms to adapt to environmental abiotic stress. Transcription factors have major roles in stress-responsive gene regulatory networks to modulate resistance to stress such as salinity and water-deficit stress [35,36]. Nuclear factor Y is a large transcription factor family. To date, many NF-Y proteins in plants have been identified and demonstrated to be involved in drought, salinity, and freezing tolerance. However, *NF-YB3* genes from CAM plants involved in abiotic stress remain obscure. In the present study, we identified the NF-Y subunit *NF-YB3* gene from the obligate CAM plant *Kalanchoe fedtschenkoi* and discovered that overexpression of it in *Arabidopsis* plants significantly improved tolerance to salinity, water-deficit stress, heat, and combined stresses in four transgenic lines. First, the seeds overexpressing *KfNF-YB3* exhibited a markedly higher germination rate than controls (WT), which showed a delayed germination phenotype under in vitro assays with mannitol, NaCl, PEG, NaCl + PEG, mannitol + PEG, and NaCl + mannitol + PEG for 16 days. This is consistent with results for *AtNF-YB3* reported by other researchers. Nelson et al. [28] found that *Arabidopsis* seeds with *AtNF-YB3* overexpression showed more vigorous germination than controls under conditions of osmotic stress. Moreover, Li et al. [34] reported a role of *AtNF-YA1* in mediating ABA-dependent seed germination and post-germination growth arrest, which is before the cotyledons turn green and develop into young seedlings under salt stress to protect the germinated seeds from water deficit.

In addition, we found that overexpression of *KfNF-YB3* in *Arabidopsis* seedlings enhanced tolerance to combined salt and water-deficit stress, which was confirmed by the longer primary root length in transgenic seedlings under NaCl and PEG treatment. Zhang et al. [37] reported that the NF-YB3 subunit from *Picea wilsonii* plays a positive role in salinity, osmotic stress, and drought stress in transgenic *Arabidospsi* at the germination stage or seedling growth stage. Previous studies have shown that root growth is closely connected with drought tolerance [38]. Ballif et al. [39] also found that overexpressing *AtNF-YB2* enhanced primary root elongation due to faster cell division and elongation. Therefore, increasing root surface area via cell elongation by *KfNF-YB3* TF might be a possible mechanism to confer combined abiotic stress response under salt and water-deficit stress.

The *KfNF-YB3* TF regulates plant growth and development. As reported by Li et al. [40], a single transcription factor may regulate multiple genes and cross talk with other TFs in a metabolic pathway. Therefore, alteration in the expression of a specific transcription factor might result in dramatic changes in growth and development, providing desired traits in the transgenic plants. Previous studies have examined the regulation of NF-YB in plant growth and development. In rice, *OsNF-YB7* has roles during both vegetative and floral meristem development. *OsNF-YB7*-overexpressing plants have a dwarf phenotype and erected leaves, as well as a dense panicle, abnormal rachis, and double flowers [41]. According to our results, *KfNF-YB3* TF regulated plant growth and development at different developmental stages. At the two-week-old mature stage, transgenic lines increased rosette size and biomass and the same pattern was observed at the five-week-old stage. In addition, *KfNF-YB3* also increased the number of leaves and the reproductive biomass (i.e., seed yield). However, transgenic lines reached similar plant heights as the WT after five weeks, even though they had significantly greater rosette size, biomass, number of leaves, and yield at the five-week-old stage. These results suggest that *KfNF-YB3* TF was regulating plant growth and development.

Some individual NFY subunits were reported to play pleiotropic regulating roles during plant growth and development. Mu et al. [42] reported the involvement of *AtNF-YA1* in male gametogenesis, embryogenesis, and seed development. These pleiotropic functions of NFYA1 might be explained by the diverse combination of NFYA1 with other NFYB/NFYC factors, which thus affected different regulating pathways. Therefore, *KfNF-YB3* TF might down-regulate the genes responsible for plant height, while up-regulating the target genes for increases in yield, biomass, rosette size, and leaves at the mature stage. Accordingly, it is interesting to gain insight into the connection of distinct *KfNF-YB3* functions, and further study on protein–protein or protein–DNA interactions is also needed to investigate the inner mechanisms.

In the current study, *KfNF-YB3* overexpression lines showed a significant acceleration in the onset of flowering, on average 10 days earlier than Col-0 WT control plants. These results indicated that, like other *NF-YB* genes, *KfNF-YB3* expression is also implicated in the regulation of reproductive organs and flowering time. Time of flowering is crucial for the reproductive success of plants and crop yield. In plants, it is coordinated by a complex network of gene regulators controlled by both environmental factors and endogenous signals [43]. The discovery of genes involved in flowering time and understanding their regulatory mechanism are important to produce crop plants with economic value. In *Arabidopsis*, the transcriptional activator *CONSTANS* (*CO)* acts as the key regulator of flowering time [44,45]. Research groups studying flowering time in both tomato (*Solanum lycopersicum*) and *Arabidopsis* identified NF-YB and NF-YC subunits as CO-interacting proteins *via* yeast two-hybrid assays [46]. Additionally, two independent research groups described flowering time delays for the *CO* mutants [47,48]. Also, Kumimoto et al. [49] found that *AtNF-YB3* mutant plants flower as late as *CO* mutants.

Furthermore, *AtNF-YB3* plays an important role in rapid flowering specifically under inductive long-day photoperiodic conditions [49]. Han et al. [50] reported that overexpression *NF-YB7* from poplar in *Arabidopsis* enhanced the development of both vegetative and reproductive organs. Chen et al. [48] found that *AtNF-YB* plays a crucial role in the regulation of flowering time in *Arabidopsis* during osmotic stress. Here, our results also suggest the function of the *NF-YB* gene in the regulation of flowering time.

*KfNF-YB3* transgenic lines exhibit improved integrated WUE. CAM plants shift their primary CO_2_ uptake and fixation to the nighttime when evaporative water losses are minimal and perform primary C_4_ and secondary C_3_ carboxylation reactions when stomata are closed during the daytime. This temporal separation of carboxylation by forming an intermediate compound called malate at night, which subsequently undergoes decarboxylation during the day, leads to reduce water loss due to transpiration [51]. Due to this capability, CAM plants achieve greater WUE than C_4_ and C_3_ plants [52,53]. Nobel [54] reported that CAM plants can fix CO_2_ 15% more efficiently than C_3_ plants and produce more biomass using less water. We have identified an *NF-YB3* gene with expression levels fourfold higher than the C_3_ state during CAM induction in older leaf pairs of the obligate CAM plant, *Kalanchoe fedtschenkoi* [14,52]. Interestingly, *KfNF-YB3* transgenic lines showed CAM-related phenotypes, such as increased biomass and WUE, at the 4-week-old mature stage, which is relevant to the higher expression of *KfNF-YB3* TF in the mature leaves of *K. fedtschenkoi*. However, the increase in WUE of transgenic *Arabidopsis* is not due to temporal separation between carbon fixation and stomatal opening but some other mechanism. To understand the connection between this CAM-related phenotype and the expression of *KfNF-YB3* TF, we need to address whether the expression level of *KfNF-YB3* increases in transgenic lines at the mature stage compared to the earlier developmental stages. Furthermore, measuring biomass and WUE in the *KfNF-YB3* knock-out mutant of *K. fedtschenkoi* would be crucial to confirm that *KfNF-YB3* has a role in increasing WUE and biomass in the CAM plant.

The KfNF-YB3 transcription factor is a highly conserved protein lacking nuclear localization signals (NLSs). This indicates a strong requirement for NF-YB/NF-YC heterodimerization for the translocation of NF-YB from the cytoplasm to the nucleus, in contrast to the NF-YA and NF-YC subunits [55]. Amino acid alignment showed that KfNF-YB3 contains conserved NF-YB sites. The conserved domain analysis of the multiple sequence alignment showed that *KfNF-YB3* is highly conserved with *AtNF-YB3* and *PwNF-YB3*. In *Arabidopsis*, *AtNF-YB3* plays an important role in promoting flowering time, especially under long-day photoperiodic conditions [49]. This aligns with the findings related to the closest orthologue of *KfNF-YB3*. Additionally, overexpression of the NF-YB3 subunit from *Picea wilsonii* conferred tolerance to salt, osmotic, and drought stress in transformed *Arabidopsis* plants and accelerated flowering [37]. These results further support our phylogenetic analysis.

For phylogenetic tree analysis, sequences of 44 homologous *NF-YB* TFs from 24 different plant species were used, predicting several functions of *KfNF-YB3* in plant growth and development, including flowering time and abiotic stress tolerance, such as salinity and drought. According to this analysis, *KfNF-YB3* clustered with an *NF-YB* TF (*AcNF-YB3*) from the edible facultative CAM plant *Ananas comosus* [56]. Although they were in two separate clusters close to each other, they shared the same hypothetical common ancestor at one internal node. Therefore, *KfNF-YB3* might play a role in shifting photosynthesis from C_3_ to CAM metabolism [57]. Further work is needed, using transcriptomic data from several CAM plants, to understand the transition from C_3_ to CAM metabolism and the underlying shifts in its regulation.

## 4. Materials and Methods

### 4.1. Phylogenetic Analysis of KfNF-YB3

Homologous sequences of *KfNF-YB3* were retrieved from the *Arabidopsis* Information Resource (TAIR, https://www.ncbi.nlm.nih.gov/) and the National Center for Biotechnology Information (NCBI) by performing a Basic Local Alignment Search Tool (BLAST) search. Sequence alignments of the *NF-Ys* were conducted using Clustal Omega software, available online: https://www.ebi.ac.uk/Tools/msa/clustalo/, accessed on 20 June 2024. Conserved motifs were investigated using DNAMAN ver. 10.3. To construct the phylogenetic tree, homologous sequences were downloaded from PlantTFDB and validated using UniprotKB. Forty-five NF-Y transcription factors (TFs) from different plant species were used to construct an unrooted phylogenetic tree using the neighbor-joining (NJ) method with MEGA10 software [58].

### 4.2. Design and Construction of the KfNF-YB3 Expression Vector

The goal of this study was to determine the multiple stress response functions of the *KfNF-YB3* gene from the obligate CAM plant *Kalanchoe fedtschenkoi* in *Arabidopsis thaliana*. To characterize the candidate CAM transcription factor, the *KfNF-YB3* gene was constitutively overexpressed in the *Arabidopsis* genome using a binary vector construct. For generation of the expression vector, the polymerase chain reaction (PCR) product of *KfNF-YB3* was cloned into an entry clone by topoisomerase I reaction via TOPO cloning (Invitrogen, Carlsbad, CA, USA). The primer pair 5′- CGT GAG CAG GAT CAT GAA GAA-3′ and 5′- TAT GAA CTC CGA AAC GCA CTC -3′ was used for PCR of the *KlNF-YB3* gene. The resultant entry clone with *KfNF-YB3* was sub-cloned into the pGWB514 binary vector [59], which has a 35S constitutive promoter and a 3×HA tag protein gene, using an LR reaction (LR Clonase, ThermoFisher Scientific, Waltham, MA).

### 4.3. Plant Material and Growth Conditions

*Arabidopsis thaliana*, ecotype Columbia (Col-0 WT), was used for all experimental and control plants in this study. All seeds were vernalized at 4 °C in the dark for four days. Seeds used for germination rate assays were sterilized using 6.1% Cl_2_ gas as described by Lindsey et al. (2017) [60] prior to plating on full-strength Murashige and Skoog (MS) basal media. All plates were kept in a plant growth chamber at 22 °C under a long-day (LD) photoperiod (16 h light/8 h dark) with a light intensity of 80 μmol m^−2^ s^−1^. After germination (approximately 5–7 days), seedlings were transplanted into individual 89 mm square plastic pots (0.3 L rooting volume, Kord, Inc., Toronto, CA) with a soil–perlite potting mixture (Scott, Miracle-Gro, USA). The plants were grown in an environmentally controlled room set at 23 °C under a 16 h day/8 h night cycle for the long-day (LD) photoperiod. All growth procedures and conditions were identical for both experimental and control plants.

### 4.4. Floral Dipping and Generation of Transgenic Lines

The final *KfNF-YB3* gene vector construct was transformed into the *Agrobacterium tumefaciens* strain GV3101 using the freeze–thaw method and the resultant plasmid was transformed into *Arabidopsis* by *Agrobacterium*-mediated floral dipping as described by Zhang et al. (2006) [61]. Seeds of T_0_ and T_1_ were screened on full-strength MS basal medium containing 75 µg mL^−1^ of hygromycin. Positive T_2_ transgenic lines were further selected with 75 µg mL^−1^ of hygromycin in MS media. Four independent homozygous T_3_ transgenic lines were used for further study using molecular assays (e.g., qRT-PCR), as well as physiological measurements including growth, reproduction (i.e., seed yield), and abiotic stress assays.

### 4.5. RNA Extraction and qRT-PCR

Total RNA was extracted from 100 mg of leaves, roots, and stem tissues of 14 *KfNFY-B3*-overexpressing lines and Col-0 WT control plants using the Quick-RNA Plant Mini Kit (Zymo Research, Irvine, CA, USA), following the manufacturer’s instructions. Quantitative real-time PCR (qRT-PCR) was performed using the Applied Biosystems QuantStudio 3 real-time PCR system (Thermo Fisher Scientific, Waltham, MA, USA) with the Luna^®^ Universal One-Step RT-qPCR Kit (New England Biolabs, Ipswich, MA, USA), according to the manufacturer’s protocol. The *KfNFY-B3* gene expression level in each transgenic line was normalized using *Actin* as the housekeeping gene and internal control. All of the primers used in gene expression analysis were synthesized by Integrated DNA Technologies, Inc. (Coralville, IA, USA). The following primer pairs were used for these analyses: *KlNF-YB3* (5′- CGT GAG CAG GAT CAT GAA GAA-3′ and 5′- TAT GAA CTC CGA AAC GCA CTC -3′) and *Actin2* (5′-CTA CGA GCA GGA GAT GGA AAC-3′ and 5′-TCT GAA TCT CTC AGC ACC AAT C-3′). Three biological replicates were used for each transgenic line and Col-0 WT plant for the qRT-PCR analysis. Relative gene expression was analyzed using QuantStudio Design and Analysis software v1.5.0 based on the comparative 2^−ΔΔCT^ method of relative gene quantification. Four selected transgenic lines from the relative gene expression analysis were used for further experiments.

### 4.6. Phenotypic Characteristics of Transgenic Lines

To determine whether *KfNFYB3* transgenic plants maintain normal productivity, phenotypic characterization was performed on third-generation transgenic lines (T_3_). Seeds of Col-0 WT plants and four transgenic lines were germinated in full-strength MS media for 5–7 days. Seedlings were then transferred to 89 mm square plastic pots containing a soil–perlite mixture (2 plants per pot, *n* = 30). Plants were grown under a long-day (LD) photoperiod (16 h light/8 h dark) at 23 °C.

Under LD conditions, rosette size and biomass of Col-0 WT control plants and transgenic lines were recorded after 5 weeks. Rosette size was measured using a sliding gauge Vernier caliper. For each plant, three measurements of rosette diameter were taken from different directions, and the average of the three measurements was recorded as the rosette size. Additionally, the plant height and number of leaves of Col-0 WT plants (*n* = 30) and four transgenic lines (*n* = 30 for each line) were recorded. Dry weights (DWs) of the plants and yield (reproductive biomass) were obtained after drying for 48 hours at 70 °C in an air oven [62]. These experiments were repeated three times.

### 4.7. Flowering Time Analysis

Seeds of *KfNF-YB3* transgenic and Col-0 WT T_3_ plants were grown under well-watered conditions in a soil–perlite potting mixture in 89 mm square plastic pots in an environmentally controlled room with a 16 h photoperiod at 23 °C. The time of appearance of the first flower buds was recorded. This experiment was repeated three times (*n* = 30 seedlings/trial).

### 4.8. Integrated Water-Use Efficiency (WUE)

Water-use efficiency (WUE) was determined as dry weight per unit of water used (mg dry weight mL^−1^ water) for four-week-old Col-0 WT and CAM TF overexpressing plants [63]. Fifty mL conical tubes were filled with a soil–perlite mixture and 35 mL of water, an amount sufficient to grow plants to the four-week-old stage. Tubes were capped with a cap that had a 2 mm diameter hole in the center. Evaporation through the hole was insignificant, especially as the plant grew and the rosette covered the hole. In each central hole, five to ten seeds were added, and the tubes were moved to a growth chamber with a 16-hour light/8-hour dark photoperiod. After germination (5–7 days), excess seedlings were removed, leaving only one seedling (*n* = 30) per tube, and each tube was weighed (W_0_). All above-cap vegetative tissue was collected after 4 weeks and dried (dry mass for each shoot = DW) for 48 h at 70 °C, and then each tube was weighed (W). Water loss was calculated as W_0_ (g) − W (g) = water used (g), where 1 g of water = 1 mL of water. Dry weight per unit of water used was calculated as DW/mL of water used. These experiments were repeated three times.

### 4.9. Water-Deficit Stress Treatment

To determine the water-deficit stress response of T_3_ transgenic plants overexpressing *KfNF-YB3*, acute water-deficit stress assays were performed. For this stress treatment, combinations of Col-0 WT control plants (*n* = 30) and *KfNF-YB3* transgenic plants (*n* = 30) were grown under well-watered conditions in a soil–perlite potting mixture in 89 mm square plastic pots for 4 weeks in an environmentally controlled room with a 16-hour photoperiod at 23 °C. After germination, the plants were irrigated well for 21 days, followed by withholding water for 15 days to induce water-deficit stress. After the 15-day stress period, the plants were re-watered for four days, and the survival rate was calculated by counting the number of green, healthy plants [64]. The dry weight (DW) of the Col-0 WT control plants and transgenic plants and yield (reproductive biomass) were obtained after drying for 48 h at 70 °C in an air oven. These experiments were repeated three times [65].

### 4.10. Heat Stress via Hypocotyl Elongation Assay

For the heat stress assay, *KfNF-YB3* TF transgenic and Col-0 WT T_3_ seeds were planted on MS agar plates, as well as on MS plates supplemented with a combination of 200 mM NaCl and −0.5 MPa PEG-8000, and 250 mM mannitol and −0.5 MPa PEG-8000 for germination. The plates were wrapped with aluminum foil to induce elongation of hypocotyls and placed vertically in a growth chamber at 22 °C for 2.5 d. The length of the hypocotyls was intended to be approximately 6–9 mm when ready for the stress treatment. The plates were then unwrapped and placed horizontally in an oven incubator set at 38 °C in the dark for a 1.5 h acclimation treatment, ensuring even heat transfer to induce heat shock proteins. During the acclimation recovery period, the plates were kept upright in the dark at 22 °C for 2 h. For the heat stress test, the plates were placed horizontally in a 45 °C incubator for 3 h. After the heat stress treatment, the plates were maintained at 22 °C for 2.5 days. Following the recovery period, the length of hypocotyl elongation was measured [66]. These experiments were repeated three times.

### 4.11. Effects of In Vitro Osmotic, NaCl, and Water-deficit Stress Treatment on Seed Germination

Sterilized seeds of *KfNF-YB3* transgenic lines and Col-0 WT plants were plated on full-strength MS basal medium supplemented with 250 mM mannitol, 200 mM NaCl, −0.5 MPa polyethylene glycol (PEG-8000), 200 mM NaCl + −0.5 MPa PEG, 250 mM mannitol + −0.5 MPa PEG, and 200 mM NaCl + 250 mM mannitol + −0.5 MPa PEG. The seeds were grown for 16 days. After 16 days, the seed germination rate was calculated. The experiment was repeated three times, with 30 plants for each line in each experiment.

### 4.12. In Vitro NaCl-Water-deficit Stress Treatment of Seedlings

For the combined NaCl and water-deficit stress assay, *KfNF-YB3* transgenic and col-0 WT control T_3_ seeds were planted on MS agar plates for germination. After 5 days of germination, seedlings (*n* = 30) from each line were transferred to new MS agar plates supplemented with 200 mM NaCl and −0.5 MPa PEG-8000 for the combined NaCl and water-deficit stress treatment. After 16 days of growth in the stress MS media, the primary root lengths of the seedlings were measured. The dry weight (DW) of the plants was obtained after drying for 48 h at 70 °C in an air oven. These experiments were repeated three times.

### 4.13. In Vitro Heat–Osmotic–NaCl Stress Treatment of 10-day-old Seedlings

*KfNF-YB3* transgenic and Col-0 WT control T_3_ seeds were planted on MS agar plates supplemented with 200 mM NaCl; 250 mM mannitol; −0.5 MPa PEG-8000; a combination of 200 mM NaCl and −0.5 MPa PEG-8000; a combination of 250 mM mannitol and −0.5 MPa PEG-8000; and a combination of 200 mM NaCl, −0.5 MPa PEG-8000, and 250 mM mannitol. The seeds were grown for 10 days under a 16-hour photoperiod at 22 °C. The plates were then placed horizontally in an oven incubator set at 38 °C in the light for a 1.5-hour acclimation treatment, ensuring even heat transfer to induce heat shock proteins. During the acclimation recovery period, the plates were kept upright at 22 °C in the dark for 2 h. For the heat stress test, the plates were placed horizontally in a 45 °C incubator for 3 h. After the heat stress treatment, the plates were maintained at 22 °C under a 16-hour photoperiod for 2.5 days. Following the recovery period, the survival percentage was determined [66]. These experiments were repeated three times.

### 4.14. Water-Deficit -Heat–Light Stress Treatment of Potted Transgenic Plants

To determine the response of T_3_ transgenic plants overexpressing the *KfNF-YB3* transcription factor to simultaneous water-deficit, heat stress, and high irradiance, a combination of Col-0 WT control plants (*n* = 30) and *KfNF-YB3* transgenic plants (*n* = 30) was grown under well-watered conditions. The plants were cultivated in soilless potting mixture in 89 mm square plastic pots for 4 weeks in an environmentally controlled room, maintained at 23 °C with a 16-hour photoperiod and 80 μmol m^−2^ s^−1^ light intensity. Water-deficit stress was induced by withholding water for six days prior to the simultaneous stress treatment. For the combined multiple stress treatment, water-deficit-stressed plants were moved to a greenhouse for three days, where temperatures can exceed 45 °C and sunlight levels can exceed 1800 µmol m^−2^ s^−1^ (Appendix A). Irradiance levels were measured using a light meter during the stress treatment. After the combined multiple stress treatment, plants were re-watered for four days, and the survival rate was calculated by counting the number of green, healthy plants. The dry weight (DW) and seed yield (reproductive biomass) of the plants were measured after drying at 70 °C for 48 h in an air oven.

### 4.15. Statistical Analysis

All data were analyzed using R v3.5.30319 [67]. Microsoft Excel vMicrosoft Office365 2019 was used to create the spreadsheets and perform Student’s t-test for pairwise comparison between two groups of samples. One-way analysis of variance (ANOVA) was performed to determine significant difference (*p* value) among multiple groups of samples of transgenic and WT plants using R [67]. One-way ANOVA analysis was done using the aov() function. F-value and associated *p*-values were calculated for each experimental dataset. Post hoc test analysis was done using the function Tukey honestly significant difference (HSD) to identify the differences in paired groups and their adjusted *p*-values for post hoc multiple paired contrasts.

## 5. Conclusions

Abiotic stresses can challenge plant growth and development even under normal environmental conditions. With the ecological impacts of climate change, these stresses are expected to become more frequent and severe [2,3]. Understanding plant responses to these intensified abiotic stresses necessitates a thorough comprehension of how plants, particularly those tolerant to multiple abiotic stresses such as CAM plants, respond under normal conditions. Our findings on the CAM-related multiple abiotic stress-responsive transcription factor response in *Arabidopsis* suggest that the abiotic stress tolerance responses of extremophytes are not necessarily unique. Supposedly, typical mesophytes and glycophytes lack individual or multiple abiotic stress response pathways. However, naturally occurring abiotic stress-tolerant genotypes within mesophytes or glycophytes have been identified using a forward genetics approach [68,69,70]. This suggests that although abiotic stress-responsive pathways are present and conserved, they are not activated or modulated appropriately [14,57,71]. Indeed, our results indicate that multiple abiotic stress responses can be modulated by an appropriate CAM-related TF to generate a multiple abiotic stress-adaptive phenotype in *Arabidopsis*, which is a glycophyte. Further research is needed to understand the molecular mechanisms responding to combined multiple stress conditions, particularly those mimicking current climate change field environments in laboratory settings. Such research will be relevant to future programs aimed at developing crop plants with enhanced resistance to increasing frequencies and severities of extreme weather events via the CAM abiotic stress-responsive transcriptional factor approach [14,71]. Additionally, such research will also provide tools to address basic questions about signaling cross talk in plant systems, especially considering that NF-Ys interact with other transcription factors [20,30,72].

## Figures and Tables

**Figure 1 ijms-25-07107-f001:**
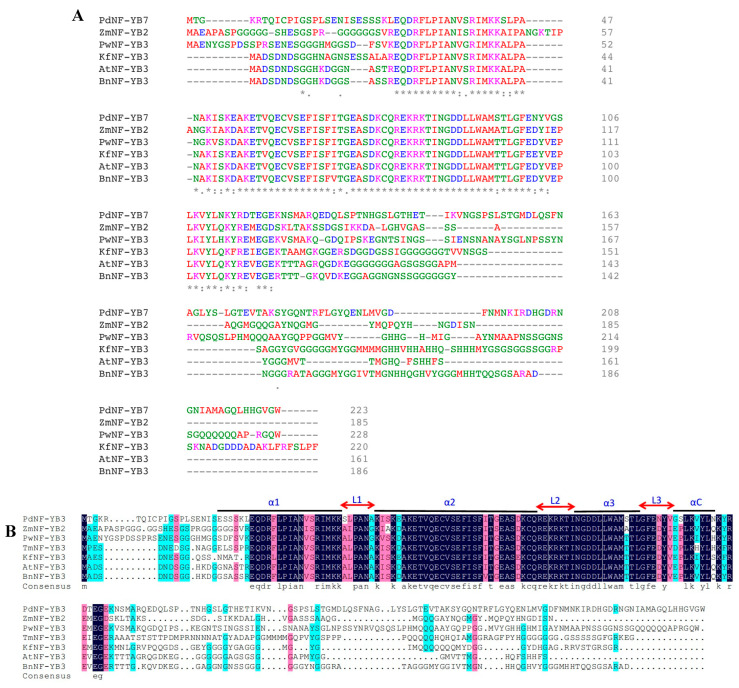
Phylogenetic analysis of the *KfNF-YB3* gene. (**A**) Multiple sequence alignment showing the conserved NF-YB binding domain among KfNF-YB3 and several abiotic-responsive NF-YB transcription factors from different plant species (* = highly conserved, : = moderately conserved, . = less conserved). (**B**) Secondary protein structures of KfNF-YB3 compared with NF-YBs from other plant species, with secondary structures presented above the alignment (helix motifs indicated by α1–αC and loops by L1–L3). (**C**) Phylogenetic tree illustrating the evolutionary relationship of *KfNF-YB3* with other abiotic stress-responsive genes in the NF-Y family.

**Figure 2 ijms-25-07107-f002:**
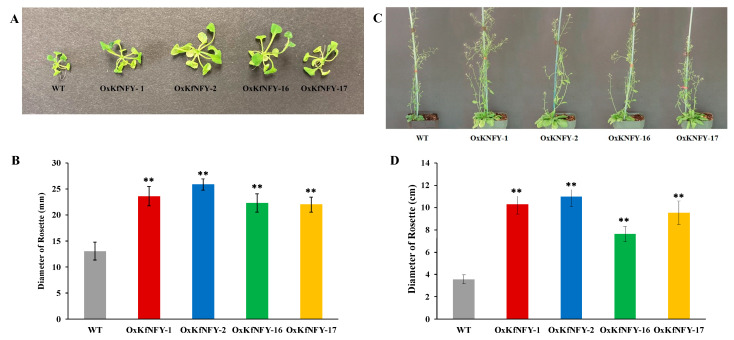
Phenotype characterization of *OxKfNF-YB3* T_3_ transgenic lines and Col-0 WT control plants. (**A**) Representative image of rosette morphology of two-week-old seedlings (*n* = 30). (**B**) Rosette size in two-week-old seedlings. (**C**) Representative image of rosette morphology of five-week-old seedlings. (**D**) Rosette size of five-week-old seedlings. Col-0 WT = gray, *OxKfNF-Y-1* = red, *OxKfNF-Y-2* = blue, *OxKfNF-Y-16* = green, *OxKfNF-Y-17* = orange. *n* = 30, experiments replicated three times, values represent means ± SE, ** adjusted *p* < 0.01.

**Figure 3 ijms-25-07107-f003:**
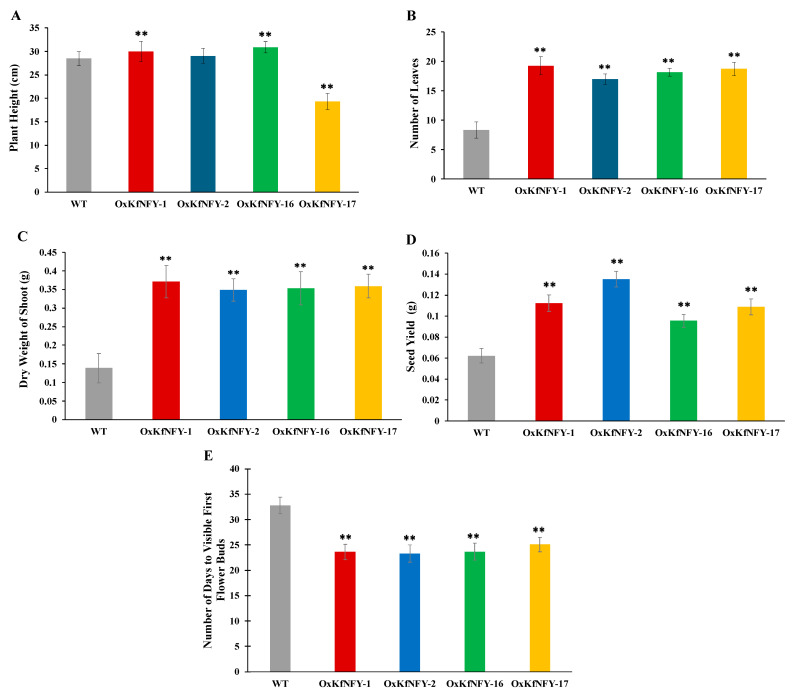
Growth characteristics of five-week-old T_3_ transgenic seedling lines and Col-0 WT control seedlings. (**A**) Plant height. (**B**) Number of leaves. (**C**) Shoot biomass. (**D**) Seed yield. (**E**) Flowering time. Col-0 WT = gray, *OxKfNF-Y-1* = red, *OxKfNF-Y-2* = blue, *OxKfNF-Y-16* = green, *OxKfNF-Y*-17 = orange. *n* = 30, experiments replicated three times, values represent means ± SE, ** adjusted *p* < 0.01.

**Figure 4 ijms-25-07107-f004:**
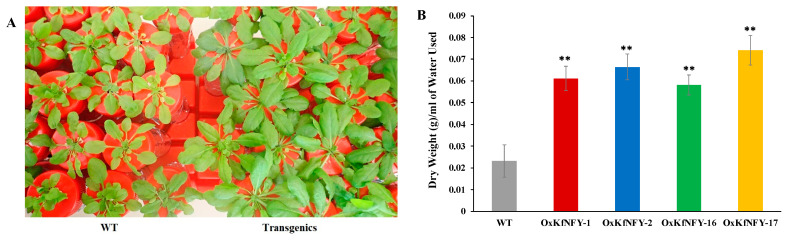
Water-use efficiency (WUE) of four-week-old *OxKfNF-YB3 T*_3_ transgenic lines and Col-0 WT control plants. (**A**) Representative image of single seedlings of transgenic lines and Col-0 WT plants after four weeks of WUE assay (*n* = 30). (**B**) Quantification of integrated WUE after four weeks. Col-0 WT = gray, *OxKfNF-Y-1* = red, *OxKfNF-Y-2* = blue, *OxKfNF-Y-16* = green, *OxKfNF-Y-17* = orange. *n* = 30, values represent means ± SE, ** adjusted *p* < 0.01.

**Figure 5 ijms-25-07107-f005:**
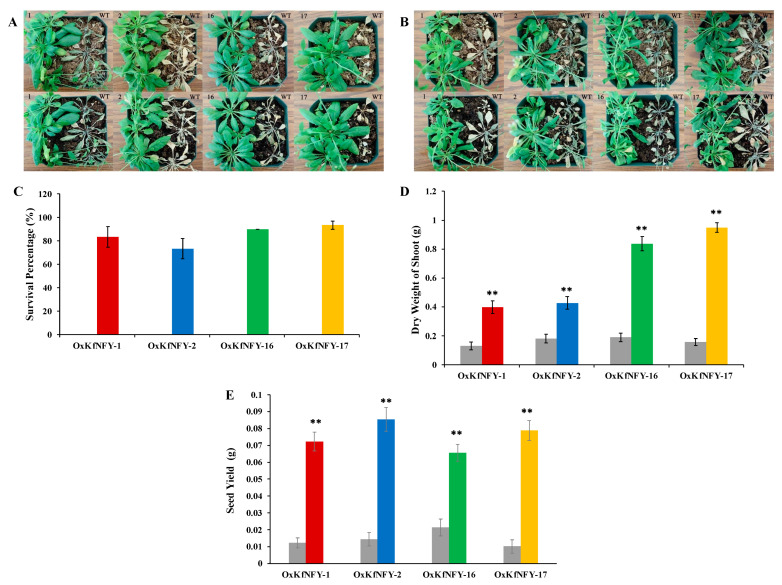
Water-deficit stress treatment of *OxKfNF-YB3 T_3_* transgenic lines and Col-0 WT control plants. (**A**,**B**) Representative image of overexpression lines and Col-0 WT control seedlings recorded at different stages during a 15-day water-deficit stress treatment (*n* = 30). (**C**) Survival rate after rewatering. (**D**) Shoot biomass after rewatering. (**E**) Seed yield after rewatering. Col-0 WT = gray, *OxKfNF-Y-1* = red, *OxKfNF-Y-2* = blue, *OxKfNF-Y-16* = green, *OxKfNF-Y-17* = orange. *n* = 10, experiments replicated three times, values represent means ± SE, ** adjusted *p* < 0.01.

**Figure 6 ijms-25-07107-f006:**
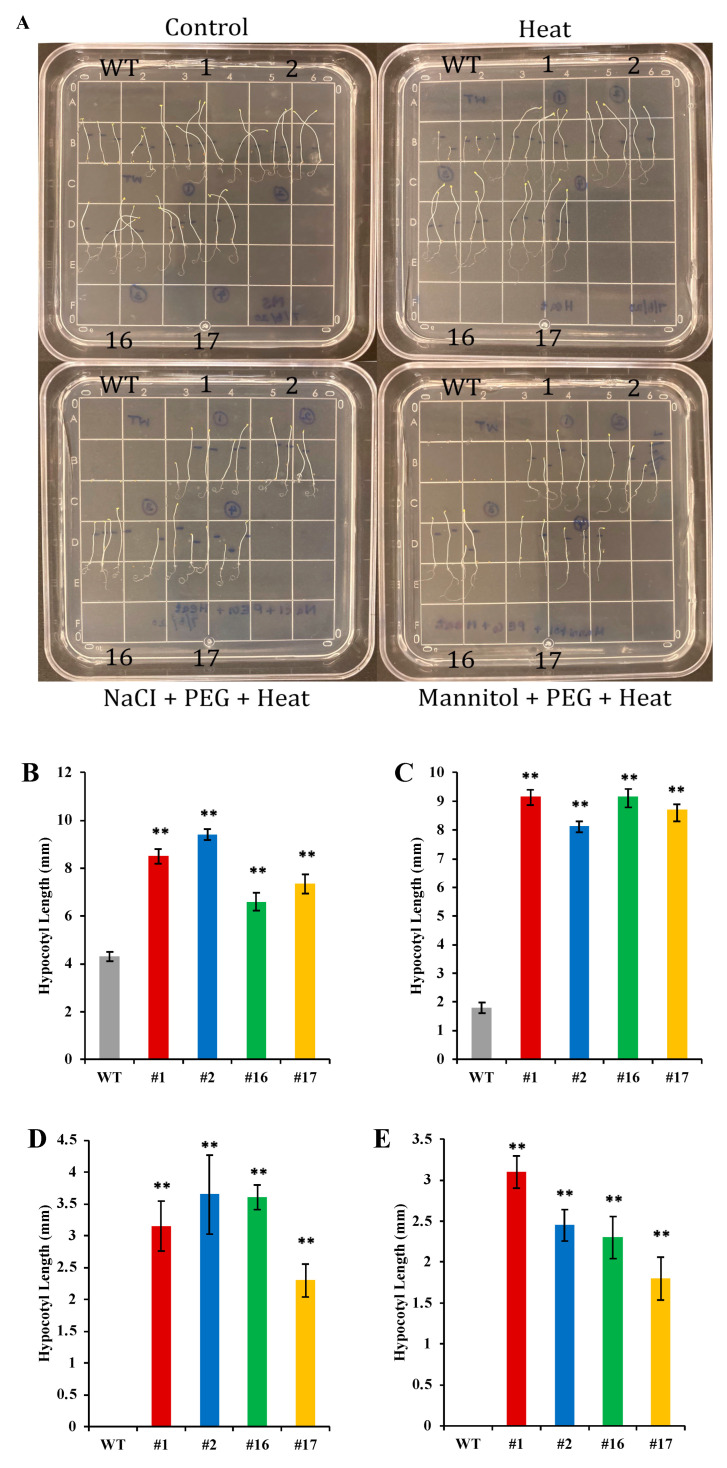
Hypocotyl length elongation in *OxKfNF-YB3* transgenic lines and Col-0 WT control plants under heat stress treatments. (**A**) Representative set up of in vitro assays of NaCl + PEG and mannitol + PEG, combined with heat stress. (**B**) No stress treatment. (**C**) Heat stress. (**D**) NaCl + PEG + heat stress. (**E**) Mannitol + PEG + heat stress. Col-0 WT = gray, *OxKfNF-Y-1* = red, *OxKfNF-Y-2* = blue, *OxKfNF-Y-16* = green, *OxKfNF-Y*-17 = orange. *n* = 30, experiments replicated three times, values represent means ± SE, ** adjusted *p* < 0.01.

**Figure 7 ijms-25-07107-f007:**
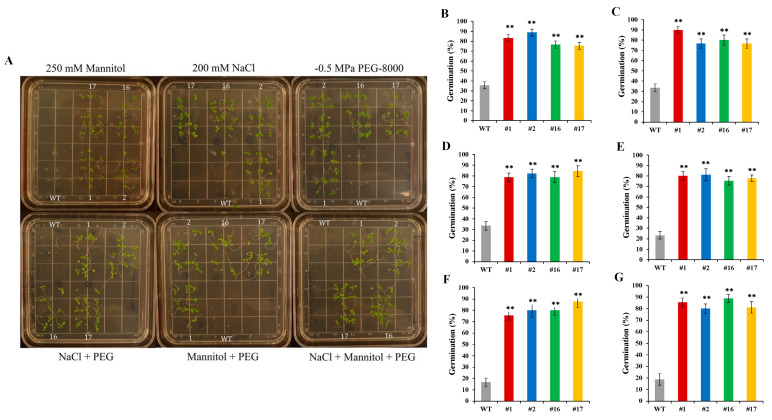
Seed germination of *OxKfNF-YB3* transgenic lines and Col-0 WT control plants under individual and combined stress treatments. (**A**) Representative set up of in vitro assays of 250 mM mannitol, 200 mM NaCl, −0.5 MPa PEG-8000, NaCl + PEG, mannitol + PEG, and NaCl + mannitol + PEG for 16 days. (**B**) NaCl stress. (**C**) PEG stress. (**D**) Mannitol stress (250 mM). (**E**) NaCl + PEG stress. (**F**) Mannitol + PEG stress. (**G**) NaCl + mannitol + PEG stress. (**H**) Combined stress assay post-germination (16 d) in MS media with 200 mM NaCl and PEG-8000. (**I**) Root length. (**J**) Biomass. Col-0 Col-0 WT = gray, *OxKfNF-Y-1* = red, *OxKfNF-Y-2* = blue, *OxKfNF-Y-16* = green, *OxKfNF-Y-17* = orange. *n* = 30, experiments replicated three times, values represent means ± SE, ** adjusted *p* < 0.01.

**Figure 8 ijms-25-07107-f008:**
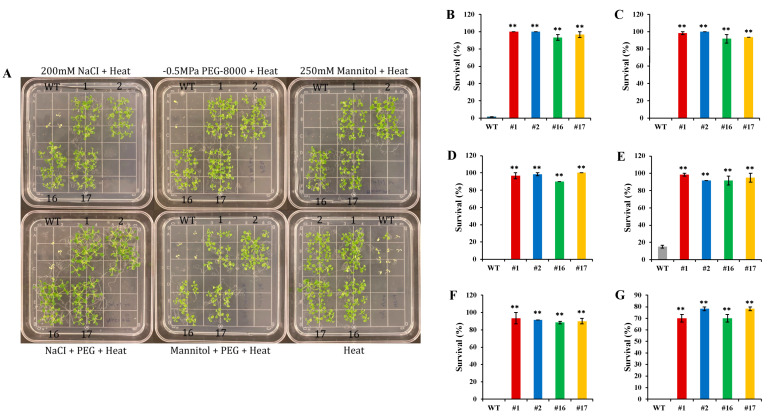
Survival rate of *OxKfNF-YB3* transgenic seedlings and Col-0 WT control seedlings under combined multiple stress treatments. (**A**) Representative set up of in vitro assays with 250 mM mannitol, 200 mM NaCl, −0.5 MPa PEG-8000, NaCl + PEG, mannitol + PEG, and NaCl + mannitol + PEG combined with heat stress. (**B**) Heat stress. (**C**) NaCl + heat stress. (**D**) Mannitol + heat stress. (**E**) PEG + heat stress. (**F**) NaCl + PEG + heat stress. (**G**) NaCl + mannitol + heat stress. Col-0 WT = gray, *OxKfNF-Y-1* = red, *OxKfNF-Y-2* = blue, *OxKfNF-Y-16* = green, *OxKfNF-Y-17* = orange. *n* = 30, experiment replicated three times, values represent means ± SE, ** adjusted *p* < 0.01.

**Figure 9 ijms-25-07107-f009:**
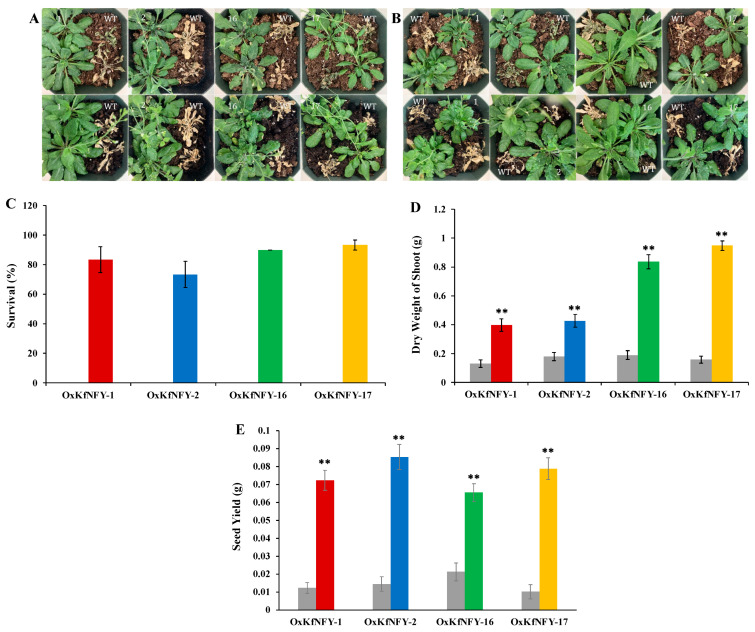
Combined multiple stress treatment with water deficit, heat, and light in *OxKfNF-YB3* T_3_ transgenic lines and Col-0 WT control plants. (**A**,**B**) Representative images of overexpression lines and Col-0 WT control plants recorded at different stages during the combined multiple stress treatment (*n* = 30). (**C**) Survival rate after rewatering. (**D**) Shoot biomass after rewatering. (**E**) Seed yield after rewatering. Col-0 WT = gray, *OxKfNF-Y-1* = red, *OxKfNF-Y-2* = blue, *OxKfNF-Y-16* = green, *OxKfNF-Y-17* = orange. *n* = 30, experiment conducted once, values represent means ± SE, ** adjusted *p* < 0.01.

## Data Availability

All data, tables, and figures in this manuscript are original and are contained within the article and Appendix A.

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
