# Peer review of "A CAM-Related NF-YB Transcription Factor Enhances Multiple Abiotic Stress Tolerance in Arabidopsis"

_ijms, 2024, doi:10.3390/ijms25137107_

Round 1

Reviewer 1 Report

Comments and Suggestions for Authors

The paper is original and interesting. The research is well conducted and resukts are sound. The only concern I have is with the presentation of the results. Most of the figures need formla imporvement to enhance the readibility and the results presentation.

Figure 1: Enlarge, Nothing can be readed.

Figures 2 and 3: delete the frames that appear in some figures and not in others. Also the ** appear with or without frame. Please delete all the frames and leave the ** alone. Some titles overlap. Please correct.

Figure 4A: Please, include different transgenic lines and depict which is which.

Figures 5 and 9: Use the same color code for the different lines that in the other figures.

Figure 6A: Please enhance the resolution and quality of the picture. Nothing can be seen.

Author Response

We appreciate your comments and suggestions for improving our manuscript. We have revised or replaced the figures with higher resolution versions as per your recommendations. However, we did not include different transgenic lines in Figure 4A because: a) they are very similar to those already depicted, b) the column chart sufficiently supports the conclusions, and c) additional images do not contribute relevant information to the conclusions.

Reviewer 2 Report

Comments and Suggestions for Authors

I have reviewed the article “Overexpression of the heat-tolerant obligate Crassulacean Acid Metabolism (CAM) plant Kalanchoe fedtschenkoi NF-YB3 transcription factor enhances combined multiple abiotic stress tolerance, water-use efficiency, growth, and development in Arabidopsis” and I understand it is valuable work in terms of the approach used and scientific contribution. I recommend it for publication but, some aspects could be improved before the acceptance.

Title: Would it be possible to summarize the title a little?

Abstract: In the abstract briefly insert the material and method.

Introduction: The introduction contextualizes the state of the art and is well-written. We only suggest the inclusion of the hypothesis, the objective, and the practical application of this type of study.

Lines 65-68: The result should not be included in the introduction: remove.

Results: Remove the methodology from the results

Lines 332-333: Is it possible to explain how this happens?

Line 546-549: There is no need to describe what Water-use efficiency is, only include the reference.

Conclusions:

Lines 653-672 The conclusion will need to be rewritten since discussions on the subject were presented and not conclusions based on the research carried out.

Author Response

We are grateful for your comments and suggestions. Below, we address each of your points. Changes in the revised version are highlighted in red in the revised manuscript.

Title: Would it be possible to summarize the title a little?

- The title has been shortened as suggested.

Abstract: In the abstract briefly insert the material and method.

- While the abstract briefly outlines the methods, we are constrained by the 200-word limit and cannot include additional details.

Introduction: The introduction contextualizes the state of the art and is well-written. We only suggest the inclusion of the hypothesis, the objective, and the practical application of this type of study.

- We have modified the introduction to incorporate your suggestion.

Lines 65-68: The result should not be included in the introduction: remove.

- It is common practice to briefly mention the main findings in the introduction, as seen in many high-impact journal articles. However, we are willing to remove the results from the introduction if this is critical for the acceptance of the manuscript.

Results: Remove the methodology from the results

- Methods mentioned in the results section have been deleted in the revised version.

Lines 332-333: Is it possible to explain how this happens?

- Further investigation is required to explain lines 332-333.

Line 546-549: There is no need to describe what Water-use efficiency is, only include the reference.

- The description of water-use efficiency has been removed in the revised version.

Conclusions:

Lines 653-672 The conclusion will need to be rewritten since discussions on the subject were presented and not conclusions based on the research carried out.

- The conclusions have been revised as suggested.

Reviewer 3 Report

Comments and Suggestions for Authors

This manuscript by Naleeka et al. found transgenic lines expressing NF-YB transcription factor gene showed faster flowering time, increased biomass, larger rosette size, higher yield and more leaves. The same transgenic lines showed higher germination rates under NaCl, osmotic and water deficit stress treatments. Through this paper, the authors dentified an NF-YB transcription factor (TF) from the heat-toler-ant obligate Crassulacean acid metabolism (CAM) plant, Kalanchoe fedtschenkoi. NF-YB transcription factor can regulator of multiple abiotic stresses. The amount of work in this manuscript is impressive, but the authors need to clarify and improve some points.

1.I think the author should refine the title, it is too long.

2.Why is there no survival rate after re-watering for WT in Figure 9C? It is hard to understand why the author did not mark the different colored columns in Figure 9C,D and E.

3.In addition, I think the data provided by the author is extremely simple. They only analyzed the transcription factor phenotypically and only obtained phenotypic results without in-depth analysis. I think it cannot match the level of IJMS, so I need the author to provide more molecular data to prove it.

Author Response

We thank you for your positive comments and your recognition of the work's significance. Below, we respond to each of your points, with changes highlighted in red in the revised version of the manuscript.

1.I think the author should refine the title, it is too long.

Response: The title has been refined as suggested in the revised version.

2.Why is there no survival rate after re-watering for WT in Figure 9C? It is hard to understand why the author did not mark the different colored columns in Figure 9C,D and E.

Response:  Regarding Figure 9C, control plants did not survive due to the combined stress from water-deficit, heat, and high light intensity, which was too severe.

Response:  Figure 9C has been revised as per your suggestion.

3.In addition, I think the data provided by the author is extremely simple. They only analyzed the transcription factor phenotypically and only obtained phenotypic results without in-depth analysis. I think it cannot match the level of IJMS, so I need the author to provide more molecular data to prove it.

Response: While molecular data to further support our conclusions is forthcoming, the current data sufficiently supports our conclusions. More importantly, one cannot “prove it” in science even with molecular data.

Reviewer 4 Report

Comments and Suggestions for Authors

In the manuscript named “Overexpression of the heat-tolerant obligate Crassulacean Acid Metabolism (CAM) plant Kalanchoe fedtschenkoi NF-YB3 transcription factor enhances combined multiple abiotic stress tolerance, water-use efficiency, growth, and development in Arabidopsis”, Naleeka R. Malwattage et al have cloned KfNF-YB3 gene and over-expressed into Arabidopsis, the transgenic lines have shown its positive function in improving salt, osmotic, and drought tolerance. The results of this research were valuable for determining its function, and helpful for genetic improvement works. However, there were some comments about it before publishing it.

Major,

(1) Authors haven’t made clear about KfNF-YB3 gene, we didn’t know how authors selected this gene, please provide essential description about its finding process.

(2) Many processes wouldn’t be described clearly, for example, primers of gene clone, gene detection, were all not claimed, please add them in supplements.

(3) All analysis about transgenic lines were focused on physiological analysis, as authors described, KfNF-YB3 genes was TF, detection of potential downstream gene expression would be helpful for determining KfNF-YB3 function.

Minor,

(4) Figure 1C, the bootstrap value should be provided, which would be easily judge phylogenetic tree reliability.

(5) In section 2.2, the common RT-PCR with gel results would be helpful for supporting positive transgenic lines, and Southern blot would be best. Meanwhile the method section would be also adjusted by adding RT-PCR gel analysis.

(6) Figure 9D and 9E have missing the legends, please add them.

(7) In section “4.15 Statistical Analysis”, the description was wrong, the statistical analysis would be performed using R platform, not R Studio. The R Studio is an operation interface, not statistical software, please support R with correct version.

(8) The sequences of KfNF-YB3 gene, including nucleotide and protein sequences, should be released, using public database or supplements.

Author Response

We appreciate your comments and suggestions. Below are our responses, with changes in red text in the revised manuscript.

(1) Authors haven’t made clear about KfNF-YB3 gene, we didn’t know how authors selected this gene, please provide essential description about its finding process.

Response: The introduction has been revised as suggested.

(2) Many processes wouldn’t be described clearly, for example, primers of gene clone, gene detection, were all not claimed, please add them in supplements.

Response: Primer information has been added in the revised version.

(3) All analysis about transgenic lines were focused on physiological analysis, as authors described, KfNF-YB3 genes was TF, detection of potential downstream gene expression would be helpful for determining KfNF-YB3 function.

Response: We agree that further analyses downstream of the KfNF-YB3 gene would be beneficial and plan to include this in our follow-up work.

(4) Figure 1C, the bootstrap value should be provided, which would be easily judge phylogenetic tree reliability.

Response: Although bootstrap values are useful for assessing phylogenetic tree reliability, our use of the tree was primarily to gain initial insights on the type of abiotic stress to apply to transgenic Arabidopsis lines. It is common practice not to report these values, as seen in molecular evolution papers (e.g., [DOI: 10.1086/666098](https://doi.org/10.1086/666098), [DOI: 10.1093/aob/mcad135](https://doi.org/10.1093/aob/mcad135)).

(5) In section 2.2, the common RT-PCR with gel results would be helpful for supporting positive transgenic lines, and Southern blot would be best. Meanwhile the method section would be also adjusted by adding RT-PCR gel analysis.

Response:  Our data support the conclusions, so we did not find it necessary to include RT-PCR gel electrophoresis.

(6) Figure 9D and 9E have missing the legends, please add them.

Response: Figures have been corrected as suggested.

(8) The sequences of KfNF-YB3 gene, including nucleotide and protein sequences, should be released, using public database or supplements.

Response: The sequence of the KfNF-YB3 gene was provided by our collaborator (as thanked in the Acknowledgments) and is unpublished. Thus, we are not in a position to release this data publicly.

Round 2

Reviewer 3 Report

Comments and Suggestions for Authors

The workload of this paper does not meet the requirements and it is recommended to reject.

Author Response

The workload of this paper does not meet the requirements and it is recommended to reject.

Response: We are uncertain about the meaning of the comment, “The workload of this paper does not meet the requirements,” and therefore, we do not have a response to this comment. However, we appreciate the reviewer’s acknowledgment that “The amount of work in this manuscript is impressive.” We have addressed and clarified the points mentioned in the resubmitted revision.

Reviewer 4 Report

Comments and Suggestions for Authors

Thanks for authors works about the revision, but some comments haven’t been addressed. For example, authors have adopted RT-qPCR to trans-genetic detection, but the error bars of these samples were big, because of high sensibility, which would be false positive. The bootstrap value is easily added in molecular evolution, but authors have remain missed them, and argued. The same problem has been in other comments. In addition, the R version has also missed in revision, and the comments was also missed in response letter.

Author Response

Thanks for authors works about the revision, but some comments haven’t been addressed. For example, authors have adopted RT-qPCR to trans-genetic detection, but the error bars of these samples were big, because of high sensibility, which would be false positive. The bootstrap value is easily added in molecular evolution, but authors have remain missed them, and argued. The same problem has been in other comments. In addition, the R version has also missed in revision, and the comments was also missed in response letter.

Response: Thank you for your comments. We appreciate that all points were addressed and that some points required rebuttal without changes to the revised version.

Regarding the qRT-PCR of transgenics, the assay was used solely to select transgenic lines, and no further inference was made from that analysis. It is evident that all four selected transgenic lines were expressing the transgene, as they survived the combined multiple abiotic assays, whereas none of the control plants did not. Our data support our conclusions, and thus we did not find it necessary to include RT-PCR gel electrophoresis. However, we have removed supplemental Figure S2 in the revised 2.0 version to focus on our strong conclusive evidence regarding the transgenic lines.

As previously mentioned, we stand by our statement regarding the bootstrap values. We are willing to remove the phylogenetic tree from the manuscript if it is critical for acceptance. Lastly, we added the R version and the citation to the revised 2.0 version. We believe these points, while noted, do not detract from the overall strong conclusions of our paper.

Round 3

Reviewer 3 Report

Comments and Suggestions for Authors

The research depth of this article is not enough for this journal. It is recommended to reject it.

Author Response

The research depth of this article is not enough for this journal. It is recommended to reject it.

Response: Thank you for your opinion. We have noted your input, but we believe the opinion, does not detract from the overall robust data supporting our conclusions in our paper.

Reviewer 4 Report

Comments and Suggestions for Authors

I have no comments about manuscript, but the manuscript should be throughtly checked, for example, the figure 2B has missed one bar.

Author Response

I have no comments about manuscript, but the manuscript should be throughtly checked, for example, the figure 2B has missed one bar.

Response: Thank you for your comments, and all figures have been checked and changed if needed in the rerevised version.